# Intrahepatic Cholestasis of Pregnancy: Neonatal Impact Through the Lens of Current Evidence

**DOI:** 10.3390/biomedicines13092066

**Published:** 2025-08-25

**Authors:** Lucia Elena Niculae, Aida Petca

**Affiliations:** 1Department of Obstetrics and Gynecology, “Carol Davila” University of Medicine and Pharmacy, 8 Eroii Sanitari Blvd., 050474 Bucharest, Romania; aida.petca@umfcd.ro; 2Department of Neonatology, Clinical Hospital of Obstetrics and Gynecology “Prof. Dr. Panait Sârbu”, 3-5 Giulesti St, 060251 Bucharest, Romania; 3Department of Obstetrics and Gynecology, Elias University Emergency Hospital, 17 Mărăști Blvd., 050474 Bucharest, Romania

**Keywords:** intrahepatic cholestasis of pregnancy, bile acids, neonatal outcomes, stillbirth, preterm birth, pregnancy complications, fetal distress, maternal–fetal medicine, early delivery, perinatal risk

## Abstract

**Background/Objectives:** Intrahepatic cholestasis of pregnancy (ICP) is the most prevalent hepatobiliary disorder unique to gestation, characterized by maternal pruritus and elevated serum bile acids. While maternal prognosis is favorable, mounting evidence links ICP to a range of neonatal complications. This narrative review aims to synthesize the current knowledge on the pathophysiological mechanisms, clinical impact and management strategies related to neonatal outcomes in ICP. **Methods:** A narrative review approach was employed, drawing on recent clinical guidelines, observational studies, mechanistic investigations and meta-analyses. Emphasis was placed on evidence exploring the relationship between maternal bile acid concentrations and neonatal morbidity, as well as on established and emerging therapeutic interventions. No systematic search strategy or formal quality appraisal was undertaken. **Results:** ICP is associated with an increased risk of adverse neonatal outcomes, including spontaneous and iatrogenic preterm birth, meconium-stained amniotic fluid, respiratory distress syndrome and stillbirth, particularly when bile acid concentrations exceed 100 μmol/L. Proposed mechanisms include placental vasoconstriction, arrhythmogenic effects and surfactant inhibition. Ursodeoxycholic acid remains the most widely used pharmacologic agent for maternal symptom relief, although evidence supporting neonatal benefit is inconclusive. Delivery by 36–37 weeks is generally recommended in cases of severe cholestasis to mitigate fetal risk. **Conclusions:** Severe ICP confers substantial neonatal risk, requiring individualized, bile-acid-guided management. While current therapies offer symptomatic maternal benefit, optimization of fetal outcomes requires timely diagnosis, vigilant surveillance and evidence-based delivery planning. Further research is warranted to refine therapeutic targets and standardize clinical practice.

## 1. Introduction

Approximately seventy years ago, Svanborg and Thorling described a recurring pattern among pregnant women who developed intense pruritus and jaundice in the third trimester, followed by spontaneous postpartum resolution. Their observations, drawn from seasonal patterns and familial clustering in Sweden, marked the first formal recognition of intrahepatic cholestasis of pregnancy (ICP) as a distinct, reversible hepatobiliary disorder of gestation [1,2].

ICP is now recognized as the most common liver disorder unique to pregnancy [3]. It typically presents in the late second or third trimester with maternal pruritus, most often affecting the palms and soles, and is biochemically defined by elevated fasting serum bile acid concentrations, frequently accompanied by abnormal liver transaminases in the absence of other liver dysfunctions [4].

The prevalence of intrahepatic cholestasis of pregnancy varies markedly by geographic region, season, and ethnicity, likely reflecting differences in genetic susceptibility and environmental factors. With a pooled global incidence of 2.9%, ICP ranges from 0.6% in Oceania to rates of 3% to 5% in parts of Asia, Europe and South America. Scandinavian women appear particularly susceptible, with population-based studies reporting incidence rates up to 3%, significantly higher than many other European cohorts, with the exception of Sweden, where a long-term nationwide analysis of more than 2 million children born between 1987 and 2010 reported a markedly lower incidence of 0.44% [5,6]. Seasonal variation has also been consistently observed, with a greater number of cases diagnosed during the winter months in temperate climates, a pattern initially described in the Nordic countries and later replicated in other populations [7].

While maternal symptoms often resolve rapidly following delivery and pose limited long-term health risks, the condition is associated with substantial perinatal morbidity [8]. Epidemiological evidence supports associations between maternal hypercholanemia and increased risks of spontaneous preterm birth, meconium-stained amniotic fluid, neonatal respiratory distress and intrauterine fetal demise, particularly when serum bile acid concentrations exceed 100 μmol/L [9,10].

Management strategies focus on maternal symptom control and fetal risk mitigation. Ursodeoxycholic acid remains the most widely used pharmacologic agent, with evidence supporting its role in lowering maternal bile acid levels and improving biochemical profiles, though its effect on fetal outcomes remains debated [11,12]. Clinical guidelines commonly recommend delivery by 37 weeks of gestation, or earlier in cases of severe cholestasis, in an effort to reduce the risk of stillbirth [8].

The emerging literature underscores the association between intrahepatic cholestasis of pregnancy and a spectrum of neonatal complications, though several aspects of the disorder remain incompletely understood. This review outlines the current understanding of its pathophysiological mechanisms, clinical presentation, and epidemiological distribution, with particular attention to the role of elevated maternal bile acids in mediating fetal and neonatal morbidity. Furthermore, therapeutic interventions, including pharmacologic management and decisions regarding the timing of delivery, are examined in the context of their effectiveness in reducing perinatal complications while highlighting key areas where further research is needed to optimize neonatal care. To assemble the evidence base, we searched PubMed and Google Scholar through July 2025 using terms such as “intrahepatic cholestasis of pregnancy”, “bile acids” and “neonatal outcomes” complemented by hand-searching the reference lists of key articles. Priority was given to recent guidelines and consensus statements, large cohort studies and mechanistic or translational work for clarifying bile acid biology in pregnancy, while classic studies of historical importance were retained to illustrate the evolution of current practice.

## 2. Pathophysiology

Bile acids are amphipathic molecules synthesized in hepatocytes from cholesterol via two primary pathways: the classical (or neutral) pathway, initiated by the enzyme cholesterol 7 alpha-hydroxylase (CYP7A1), and the alternative (or acidic) pathway, initiated by mitochondrial sterol 27-hydroxylase (CYP27A1)—see Figure 1. In addition, two minor routes of bile acid synthesis have been described: a 25-hydroxylase pathway active in the liver and peritoneal macrophages, and a 24-hydroxylase pathway occurring exclusively in the brain, in neurons and astrocytes [13,14]. The classical pathway is predominant in humans and produces the primary bile acids (cholic acid and chenodeoxycholic acid) which are then conjugated to either glycine or taurine to improve solubility. Conjugated bile acids are actively secreted into the bile canaliculi by the bile salt export pump (BSEP), encoded by the *ABCB11* gene, and are ultimately stored in the gallbladder. After food intake, they are released into the duodenum, where they emulsify dietary fats and facilitate lipid and fat-soluble vitamin absorption. Approximately 95% of bile acids are reabsorbed in the terminal ileum by the apical sodium-dependent bile acid transporter (ASBT), returned to the liver via the portal vein, and taken up by hepatocytes through sodium taurocholate cotransporting polypeptide (NTCP) and organic anion transporting polypeptides (OATPs), thus maintaining an efficient enterohepatic circulation [14,15].

Bile acid synthesis is tightly regulated through a negative feedback loop governed by the farnesoid X receptor (FXR), a nuclear receptor activated by intracellular bile acids. In the liver, FXR downregulates CYP7A1 to limit further bile acid synthesis, while in the ileum, it induces the production of fibroblast growth factor 19 (FGF19), which acts hormonally to further suppress hepatic bile acid synthesis. Together, this hepatic–intestinal axis helps maintain bile acid homeostasis under physiological conditions [16].

In addition to FXR, bile acids also signal through the Takeda G protein-coupled receptor 5 (TGR5), a membrane-bound receptor expressed in extrahepatic tissues such as enterocytes, gallbladder epithelium, adipose tissue, and immune cells. While TGR5 does not directly regulate bile acid synthesis, it modulates systemic bile acid responses by promoting glucagon-like peptide-1 secretion, gallbladder relaxation, energy expenditure, and anti-inflammatory effects [13,14].

## 3. Etiology

The underlying cause of ICP is not fully defined, but current evidence points to a multifactorial process involving genetic susceptibility, placental remodeling, hormonal and microbiome disruption, immune imbalances, environmental exposures, iatrogenic triggers and emerging molecular signatures identified through proteomic and metabolomic studies [17]. Rather than acting in isolation, these factors interact across physiological systems to impair bile acid regulation and contribute to disease expression in susceptible pregnancies.

### 3.1. Genetic Factors

Genetic predisposition plays a central role in the development of intrahepatic cholestasis of pregnancy, with multiple studies identifying pathogenic or functionally relevant variants in genes involved in bile acid transport and hepatocellular function. The most consistently implicated gene is *ABCB4* (ATP-binding cassette, subfamily B, member 4), which encodes multidrug resistance protein 3 (MRP3), a canalicular phospholipid floppase essential for the formation of protective bile micelles. Heterozygous variants in *ABCB4* have been identified in up to 15–20% of women with ICP, supporting the idea of a dose-sensitive impairment in phospholipid secretion that compromises bile fluidity and increases cholestatic risk [18]. Mutations in *ABCB11*, encoding the bile salt export pump, have also been reported, particularly in severe, recurrent or early-onset cases. Additional candidate genes include *ATP8B1*, involved in maintaining canalicular membrane integrity, *TJP2*, encoding a tight junction protein associated with cholestatic syndromes and *NR1H4*, which encodes the nuclear farnesoid X receptor. While no single variant appears causative in isolation, the combined effect of several polymorphisms may lower the threshold for cholestasis when superimposed on the metabolic and hormonal milieu of pregnancy [7,19,20].

Further supporting a genetic contribution are observations of familial clustering, with increased prevalence of ICP among first-degree female relatives, including mother–daughter and sister pairs. Twin studies suggest heritability as well, with higher concordance in monozygotic than dizygotic twins [21]. Moreover, marked geographic and ethnic variability, such as the high incidence reported in Chilean Mapuche populations and certain Scandinavian regions, underscores the role of population-specific genetic background in modulating susceptibility. These patterns, unlikely to result solely from environmental exposures, reinforce the hypothesis that inherited defects in bile acid handling represent a key component in ICP pathogenesis [22].

### 3.2. Hormonal Factors

Steroid hormones, particularly progesterone and estrogen, play a pivotal role in the pathogenesis of ICP, not through their overall circulating levels, which are physiologically elevated in all pregnancies, but rather through the accumulation of specific cholestatic metabolites and altered receptor interactions that impair bile acid homeostasis. Among these, sulfated progesterone metabolites such as epiallopregnanolone sulfate and allopregnanolone sulfate have been shown to exert inhibitory effects on the farnesoid X receptor. By attenuating FXR signaling and trans-inhibiting BSEP at the canalicular membrane, these metabolites disrupt bile acid efflux and promote intrahepatic retention, leading to a cholestatic phenotype in genetically susceptible women [23]. Additional experimental work has demonstrated that they can also inhibit NTCP, further restricting bile acid uptake and clearance, thereby amplifying the hepatic burden of bile acids [24].

Estrogen contributes to this process through complementary, but distinct mechanisms. Elevated concentrations of 17β-estradiol during late gestation suppress BSEP transcription via a non-classical interaction between estrogen receptor α (ERα) and FXR, reducing transporter expression at the hepatocyte canalicular surface [25]. Moreover, estrogen upregulates microRNA-148a, which downregulates pregnane X receptor signaling and alters MRP3 activity, thereby impairing compensatory bile acid efflux into the systemic circulation. These molecular effects converge to diminish hepatobiliary bile acid clearance, explaining why ICP often manifests in the third trimester, when hormone levels peak and their cholestatic metabolites accumulate [26].

In rodent models, administration of sulfated progesterone metabolites reproduces a cholestatic phenotype, characterized by hypercholanemia and reduced FXR target gene expression [27], while estrogen-treated mice display transcriptional repression of BSEP and canalicular dysfunction that mirror the biochemical and clinical features of ICP [28].

### 3.3. Placental Factors

Far from being a passive conduit between mother and fetus, the placenta is a dynamic endocrine and transport organ whose adaptive capacity is crucial to maintaining homeostasis during pregnancy. In the context of intrahepatic cholestasis of pregnancy, this adaptive machinery is disrupted, leading to impaired detoxification of maternal bile acids and heightened fetal exposure. Mounting evidence indicates that the placenta in ICP is not simply overwhelmed by excess bile acids, but rather exhibits specific structural and molecular alterations that compromise its ability to act as an effective barrier [29].

A central feature of this dysfunction is the dysregulation of bile acid transport systems. The expression of transporters such as OATP1A2 and OATP1B3 is markedly reduced in ICP placentas, restricting the clearance of bile acids from the fetal to maternal circulation. Although some compensatory upregulation of alternative efflux pathways has been described, these adaptations are insufficient, particularly in severe cases, resulting in persistent accumulation of toxic bile acids on the fetal side [30].

Beyond transporter expression, ICP placentas display distinct structural and cellular abnormalities. Histological analyses reveal reduced intervillous space, an increased prevalence of syncytial knots and enhanced capillary growth within terminal villi. These structural signatures, compatible with chronic low-grade hypoxia, are thought to arise from bile-acid-induced vasoconstriction within the intervillous circulation [31,32].

Finally, upregulation of enzymes such as SRD5A1 and AKR1C2 enhances local production of sulfated progesterone metabolites, which not only reinforce maternal FXR inhibition, but also impair placental transport functions, creating a feedback loop in which altered steroid metabolism and bile acid toxicity perpetuate one another [33].

Taken together, these abnormalities delineate a characteristic “placental phenotype” of ICP, defined by impaired bile acid transport, structural remodeling consistent with hypoxic stress and altered placental steroid metabolism, which collectively reflect the inability of the placenta to adapt to the cholestatic environment [31].

### 3.4. Immunological Factors

Immune dysregulation is increasingly recognized as a relevant contributor to the pathophysiology of intrahepatic cholestasis of pregnancy. In a normal uncomplicated pregnancy, the maternal immune system undergoes a tightly regulated shift toward a predominantly anti-inflammatory Th2-profile, which promotes tolerance to the semi-allogeneic fetus. In contrast, this immunologic balance appears to be disturbed in ICP, with evidence pointing to a shift toward a Th1 pro-inflammatory response. This is reflected in elevated levels of cytokines such as tumor necrosis factor-alpha (TNF-α), interferon-gamma (IFN-γ), interleukin-6 (IL-6) and interleukin-12 (IL-12), alongside decreased levels of interleukin-4 (IL-4) and transforming growth factor-beta 2 (TGF-β2), affecting the maternal–fetal immune tolerance and promoting systemic inflammation [34,35,36].

Within the placenta, increased activation of the nuclear factor kappa B (NF-κB) pathway has been observed, particularly in trophoblast cells exposed to high bile acid levels. This pathway enhances the production of inflammatory mediators and may contribute to local tissue injury. At the same time, the expression of peroxisome proliferator-activated receptor gamma (PPARγ), which normally counteracts NF-κB activity and maintains an anti-inflammatory environment, appears to be reduced in ICP, further amplifying placental inflammation [37,38].

Finally, disruption of autophagy, an intracellular degradation and recycling pathway that regulates immune signaling, has emerged as a potential link to the development of the disease. In both hepatocytes and placental cells, impaired autophagic activity has been associated with bile acid accumulation, oxidative stress and excessive cytokine production, associating cellular stress responses to systemic immune activation in ICP pregnancies [39,40].

### 3.5. Microbiome Factors

In the course of an uncomplicated pregnancy, the maternal gut microbiome undergoes profound, but physiological restructuring. This process is characterized by a gradual decline in microbial diversity and the expansion of metabolically active taxa that enhance maternal energy harvest and adapt nutrient processing to meet the demands of the growing fetus. Among these, Bacteroidetes species display increased bile salt hydrolase activity, leading to enhanced deconjugation of bile acids in the intestinal lumen. This reduces their reabsorption by the apical sodium-dependent bile acid transporter in the ileum and attenuates FXR activation, thereby subtly modifying the enterohepatic circulation of bile acids [41,42].

However, in ICP, this finely tuned adaptation appears to be disturbed. Recent studies have reported distinct microbial signatures in affected women, including further reductions in microbial diversity and a selective proliferation of species such as Bacteroides fragilis. These alterations have been linked to impaired enterohepatic signaling and abnormal bile acid metabolism. As a consequence, the bile acid pool shifts toward more hydrophobic and hepatotoxic species, exacerbating hepatic retention and amplifying cholestatic injury in genetically predisposed individuals [41,43].

Taken together, these observations highlight the gut–liver axis as an important determinant of bile acid homeostasis in pregnancy and suggest that microbial dysbiosis in ICP may represent not merely a secondary effect, but an active driver of disease pathogenesis [32,43].

### 3.6. Environmental Factors

Given the marked seasonal variation in ICP incidence observed in regions such as Scandinavia and Chile, attention has turned toward environmental and dietary exposures that may influence disease risk. Significantly lower serum selenium levels and reduced glutathione peroxidase activity have been well documented in affected pregnancies compared to healthy controls, particularly during winter months, indicating impaired antioxidant capacity potentially linked to cholestasis [44,45]. Similarly, vitamin D deficiency has been associated with ICP as pregnant women diagnosed with intrahepatic cholestasis of pregnancy exhibit lower serum 25-hydroxyvitamin D concentrations, which may impact bile acid regulation and immune function. Environmental factors, such as reduced ultraviolet B exposure and increased ambient air pollution (notably higher fine particulate matter), during pregnancy or in the periconceptional period, further impair vitamin D synthesis and have been correlated with higher ICP risk in large cohort studies [46].

### 3.7. Iatrogenic Factors

Certain pharmacologic agents have been associated with cholestasis during pregnancy, particularly those that affect hepatobiliary transport or hormone metabolism. Among them, assisted reproductive protocols involving high-dose estradiol or progesterone have long been recognized for their cholestatic potential. Estrogens can inhibit the expression and function of the bile salt export pump and reduce bile acid clearance, mechanisms that parallel those implicated in ICP pathogenesis [47]. Additionally, some antibiotics (azithromycin, amoxicillin) and proton pump inhibitors (lansoprazole, omeprazole) have been reported to induce cholestatic liver injury when used in susceptible individuals. However, evidence remains limited and causality is often difficult to establish in the context of underlying genetic predisposition [48,49].

### 3.8. Proteomics and Metabolomics

Recent omics-based research has identified distinct molecular profiles in intrahepatic cholestasis of pregnancy. Metabolomic analyses consistently report elevated levels of conjugated bile acids, along with disruptions in amino acid metabolism and increased long-chain saturated fatty acids, all correlated with disease severity and lower antioxidant capacity [50]. Furthermore, proteomic profiling identified proteins involved in inflammation, lipid metabolism and mitochondrial function that differ significantly in ICP, enabling discrimination from healthy pregnancies with high accuracy [51]. Notably, ACOX1, L-palmitoylcarnitine and glycocholic acid have emerged as a potential biomarker panel for early detection [52]. Collectively, these signature profiles reflect broader disturbances in bile acid handling, oxidative stress and immune activation, providing promising targets for early diagnosis and tailored therapies.

## 4. Clinical Presentation and Diagnosis

The hallmark symptom of intrahepatic cholestasis of pregnancy is maternal pruritus, which typically arises in the third trimester, although it may present earlier in the late second trimester. The itching classically affects the palms and soles and tends to intensify at night, often progressing to involve other areas of the body. Unlike other dermatoses of pregnancy, pruritus in ICP occurs in the absence of primary skin lesions and is not associated with systemic illness [53]. In most cases, symptoms resolve rapidly within days postpartum, a feature that further supports the diagnosis. Jaundice develops in few cases, while other nonspecific symptoms, such as dark urine, steatorrhea, right upper quadrant discomfort and fatigue, may occur, but are less common and less specific [54,55].

Diagnosis of ICP is based on the presence of characteristic pruritus and biochemical evidence of cholestasis, most notably an elevated serum total bile acid concentration. While the threshold is generally considered to be a serum bile acid concentration of 10 μmol/L or greater, international guidelines differ slightly in their recommendations regarding sampling and cut-off values. The American Gastroenterological Association advises using the recommended threshold and further notes that serum bile acid levels may be physiologically elevated during pregnancy and that testing should be performed in the non-fasting state to avoid underestimation [4]. Similarly, the European Association for the Study of the Liver accepts a non-fasting total bile acid level of 10 μmol/L or greater in the presence of pruritus as sufficient for diagnosis [56]. In contrast, the UK, Australian and Canadian recent guidelines recommend a higher threshold of 19 μmol/L or greater for non-fasting samples, recognizing the influence of postprandial bile acid fluctuations. However, beyond their diagnostic role, total serum bile acid concentrations are also used to stratify disease severity and guide clinical decision-making, with all major international guidelines endorsing a uniform classification system. Mild ICP is generally defined by bile acid levels up to 39 μmol/L, moderate disease by concentrations between 40 and 99 μmol/L and severe ICP, where the risk of adverse perinatal outcomes appears to increase substantially, by levels greater than 100 μmol/L [57]. In cases where initial bile acid levels are normal but clinical suspicion remains high, repeat testing after one to two weeks is advised, as bile acid concentrations may rise with advancing gestation [58,59,60]. Other laboratory findings include moderately elevated liver transaminases up to 10–20 times their normal value, which may precede the rise in bile acid levels. Finally, alkaline phosphatase is typically elevated in all pregnancies due to placental production and is not specific for ICP [55].

The diagnosis of intrahepatic cholestasis of pregnancy requires exclusion of other hepatic or systemic conditions that may present with pruritus or abnormal liver function tests during gestation. Likely causes include viral hepatitis, often presenting with systemic symptoms (fever, malaise, anorexia, nausea, abdominal pain) and more pronounced jaundice, which are easily diagnosed by serologic testing for viral markers. Also characterized by severe nausea, hyperemesis gravidarum can cause mild transaminase elevation, but is not usually associated with isolated pruritus. Additionally, chronic liver diseases such as autoimmune hepatitis or primary biliary cholangitis are rare in pregnancy and can be excluded through specific autoantibodies. Gallstone disease is another differential diagnosis for ICP, presenting with right upper quadrant pain, jaundice and cholestatic liver enzyme pattern, the diagnosis being supported by imaging. Also, drug-induced liver injury should be considered when there is a history of exposure to hepatotoxic medications or supplements, with a variable pattern of liver test abnormalities [55].

Hypertensive disorders of pregnancy, particularly those with hepatic involvement, must also be carefully considered in the differential diagnosis. Acute fatty liver of pregnancy (AFLP) and HELLP syndrome are critical to exclude in pregnant patients presenting with abnormal liver tests, especially in the third trimester. AFLP often presents with nonspecific symptoms such as nausea, vomiting, abdominal pain and malaise, and is typically accompanied by marked transaminase elevation, hyperbilirubinemia, hypoglycemia, leukocytosis and coagulopathy. For diagnosis and assessing the severity of disease, the Swansea criteria are recommended, although the condition remains one of exclusion and clinical judgment. HELLP syndrome, defined by hemolysis, elevated liver enzymes and low platelet count, is another important differential, usually occurring in the setting of preeclampsia. It frequently coexists with hypertension and proteinuria and is characterized by right upper quadrant or epigastric pain, elevated lactate dehydrogenase and features of microangiopathic hemolytic anemia. Both AFLP and HELLP require urgent obstetric management and are generally distinguishable from ICP by their acute presentation, systemic involvement and laboratory profile [4,56,61]. A summary of the recommended management of ICP is provided in Figure 2.

Overall, the diagnosis of ICP rests on the integration of clinical presentation with supportive biochemical findings, particularly elevated serum bile acids. Ongoing monitoring is recommended, as bile acid levels may fluctuate, and higher concentrations are associated with increased fetal risk. Prompt recognition and accurate diagnosis are essential for guiding management and reducing perinatal complications.

## 5. Neonatal Outcomes Associated with ICP

### 5.1. Preterm Birth

Preterm birth is one of the most frequently documented neonatal complications associated with intrahepatic cholestasis of pregnancy, with growing evidence supporting both the clinical association and its mechanistic pathways. A recent dose–response meta-analysis, based on a large cohort of over 15,800 women, demonstrated that even moderate elevations in serum bile acid concentrations (20–39 µmol/L) are associated with increased odds of spontaneous preterm birth, while levels 40 µmol/L and over markedly elevate the combined risk of both spontaneous and iatrogenic preterm delivery [57]. Complementing this, another meta-analysis encompassing more than 1.5 million women similarly reported a strong association between ICP and preterm birth, showing progressively higher risk corresponding to rising disease severity [62].

In humans, elevated bile acid concentrations enhance the expression of myometrial oxytocin receptors, thereby increasing uterine contractility and triggering spontaneous preterm labor [63,64]. In parallel, bile acids (particularly cholic acid) exert dose-dependent vasoconstrictive effects on placental chorionic veins, contributing to vascular dysfunction and fetal distress, which can be associated with iatrogenic preterm delivery [65]. These mechanisms are coupled with an inflammatory environment activated by high levels of serum bile acids, through the NF-κB pathway via G protein-coupled bile acid receptor 1 (Gpbar1), leading to aberrant leukocyte infiltration in the placenta [66]. Animal models reinforce these findings, showing that elevated bile acids alone are sufficient to trigger preterm labor in the absence of other obstetric complications [67,68].

### 5.2. Fetal Distress and Meconium-Stained Amniotic Fluid

Following the elevated risk of preterm birth, fetal distress manifesting as meconium-stained amniotic fluid (MSAF) is increasingly recognized as one of the adverse neonatal outcomes particularly associated with severe or early-diagnosed ICP. With an incidence ranging from 15% to 48%, the risk of MSAF is significantly higher in women reaching serum bile acid levels ≥100 μmol/L and is further amplified in early-onset severe ICP, a trend demonstrated by two large retrospective cohorts in the Dutch and Argentinian population, respectively [69,70].

This is primarily due to elevated maternal bile acids crossing the placenta and increasing fetal serum bile acid concentrations. Consequently, this stimulates the fetal enteric nervous system, leading to increased gut motility and premature relaxation of the anal sphincter, resulting in intrauterine passage of meconium [71]. Evidence for the involvement of bile acids in the etiology of meconium staining of the amniotic fluid comes from in utero studies dating back to the 1970s in rabbits and later lambs where induced hypercholanemia provoked bradycardia and meconium passage, even in the absence of infection or uterine contractions [67,72,73].

The American Association for the Study of Liver Diseases and the Society for Maternal-Fetal Medicine both emphasize that the presence of meconium-stained amniotic fluid in ICP is not a benign finding and represents a signal for acute fetal compromise. Therefore, continuous fetal monitoring becomes imperative and timely delivery is often initiated to mitigate the risk of meconium aspiration and neonatal respiratory distress [8,74].

### 5.3. Fetal Demise

Following the spectrum of neonatal complications, intrauterine fetal demise (IUFD or stillbirth) represents the most concerning outcome linked to intrahepatic cholestasis of pregnancy. Although stillbirth remains uncommon in mild cases, robust evidence indicates a sharply elevated risk when maternal serum bile acid concentrations exceed 100 µmol/L. Based on a recent meta-analysis published by Ovadia and colleagues, including a cohort of approximately 170,000 pregnant women, the prevalence of IUFD reaches 3.4% in severe ICP cases, compared with 0.3% in the general obstetric population, a more than tenfold increase underscoring the possible bile acid toxicity [10].

The mechanism through which fetal demise occurs in the setting of ICP is not completely understood. However, there has been strong evidence supporting the arrhythmogenic effect of bile acids on the fetal heart, as multiple case reports have described fetal atrial flutter and supraventricular tachycardia in cholestatic pregnancies [75,76,77]. This is substantiated by studies correlating high bile acid concentrations with markers of fetal cardiac dysfunction, such as increased NT-proBNP, PR interval prolongation and altered heart rate variability [78]. In parallel, in vitro studies of rat fetal cardiomyocytes exposed to taurocholic acid reveal reversible disturbances in rhythm and contractility, but irreversible damage linked to tauro conjugate accumulation, reinforcing a mechanistic link between ICP and fetal myocardial injury [79,80].

The fetal myocardial performance index (MPI), particularly the left ventricular modified MPI (LMPI), has been studied as a marker of fetal cardiac dysfunction in intrahepatic cholestasis of pregnancy. Multiple echocardiographic assessments of fetuses exposed to severe ICP consistently display impaired left ventricular functions, as evidenced by elevated LMPIs, with reduced systolic strain and prolonged isovolumetric contraction and relaxation times. Nevertheless, the low reproducibility of MPI measurements and the lack of standardized cut-off values limit its application in routine clinical practice as an early surveillance tool [81,82].

In aggregate, these data reveal a pathophysiological continuum wherein elevated bile acids cause myocardial stress, electrophysiological instability and impaired contractile performance. This cascade may culminate in sudden fetal arrhythmia or circulatory collapse, leading to stillbirth, a trajectory often occurring without warning in late gestation. Consequently, expert guidelines now recommend intensified fetal monitoring when serum bile acids ≥ 100 µmol/L, with expedited delivery when cardiac compromise is identified [4,8,56].

### 5.4. Low Birth Weight and Small for Gestational Age

The impact of intrahepatic cholestasis of pregnancy on fetal growth remains an area of ongoing debate, with conflicting evidence reported across populations and study designs. While some data suggest a potential association between ICP and impaired fetal growth, other authors report no significant effect, while a subset even indicates a tendency toward large for gestational age infants.

Several large-scale studies, particularly from East Asian populations, have identified an increased risk of small for gestational age (SGA) and low birth weight (LBW) among ICP pregnancies. In a retrospective cohort of over 11,800 singleton deliveries in China, Li et al. found a significant increase in SGA rates among women meeting clinical diagnostic criteria for ICP [83]. In a similar setting, Song and co-authors conducted a population-based analysis of more than 68,000 singleton pregnancies and reported an increased risk of both SGA and LBW in women with ICP, with the magnitude of risk rising proportionally with maternal bile acid levels [84]. In contrast, other studies conducted in Western populations have not observed an independent association between ICP and impaired fetal growth, a view that aligns with the one adopted by major clinical guidelines [8,85]. Adding further complexity, a large Swedish retrospective cohort of over 1.2 million singleton deliveries reported an increased incidence of LGA infants among ICP pregnancies compared to the general population, raising the possibility of bile-acid-associated metabolic alterations promoting excess fetal growth in some settings [86].

Variability in study populations, bile acid thresholds, disease severity, maternal characteristics and timing of delivery may all contribute to the observed heterogeneity in reported fetal growth outcomes in ICP. However, the overall consensus remains that most deviations in birth weight reflect factors indirectly associated with intrahepatic cholestasis of pregnancy, such as premature birth, rather than intrinsic growth restriction. As such, individualized surveillance may be considered in selected cases, but routine monitoring is not universally indicated. Current guidelines recommend regular measurement of maternal serum bile acid levels primarily to guide decisions on delivery timing and to reduce the risk of stillbirth and other severe perinatal outcomes, rather than to specifically prevent low birth weight [4].

### 5.5. Neonatal Respiratory Distress Syndrome

Neonatal respiratory distress syndrome (RDS) is a recognized complication in newborns of mothers with intrahepatic cholestasis of pregnancy, with a prevalence reported as high as 17–29% in case–control and cohort studies, significantly exceeding the rates in the general population even after adjustment for gestational age and mode of delivery. The association between ICP and neonatal RDS was first described in the early 2000s as “bile acid pneumonia”, with initial case series and observational studies documenting severe RDS in near-term infants born to mothers with ICP, despite evidence of lung maturity (lecithin/sphingomyelin ratio greater than 2), while identifying high bile acid levels in both neonatal serum and bronchoalveolar lavage fluid [9,87,88,89].

The proposed pathophysiological mechanism centers on the transplacental passage of bile acids, which accumulate in the fetal lung and disrupt surfactant function. In healthy pregnancies, the fetal hepatobiliary system synthesizes a compositionally distinct, small bile acid pool via the alternative pathway. Owing to the immaturity of the fetal enterohepatic circulation, fetal bile acid clearance depends predominantly on placental transfer to the maternal circulation, where maternal hepatic metabolism and biliary excretion maintain fetal–maternal bile acid homeostasis. This balance is tightly regulated by placental expression of specialized transporters (such as OATP3A1, OATP4A1) and by the apical sodium-dependent bile acid transporter, together preserving a low fetal-to-maternal bile acid gradient. However, in intrahepatic cholestasis of pregnancy, maternal bile acid levels rise due to impaired hepatic clearance, saturating the placental transport systems and reversing the physiological gradient. As a result, excess bile acids accumulate in the fetal circulation and are delivered to peripheral tissues, including the lung [90,91]. Although the possibility of bile acid aspiration from the amniotic fluid has been proposed, this mechanism is not strongly supported, as amniotic bile acid levels do not correlate with neonatal morbidity and meconium aspiration is frequently absent in cases with bile acid levels lower than 100 μmol/L [92].

As contributors to the pathogenesis of neonatal respiratory distress syndrome, bile acids are thought to induce surfactant depletion by activating phospholipase A2, leading to degradation of phosphatidylcholines and impaired alveolar stability, as well as triggering inflammatory responses via macrophage activation [9,93]. In addition to bile-acid-mediated surfactant dysfunction, acute fetal placental hypoxia has also been implicated through several interrelated molecular pathways. Dysregulation of vasoactive mediators, including reduced placental and maternal levels of urocortin- and corticotropin-releasing hormone, impairs utero placental perfusion [94,95]. Also, upregulation of hypoxia inducible factor 1α and downregulation of inducible nitric oxide synthase further exacerbate ischemia-reperfusion stress, promoting fetal hypoxia and worsening pulmonary vulnerability [96,97].

These findings are corroborated by animal studies, particularly in swine and rabbits, respectively, where severe impairment of gas exchange and lung mechanics was observed [98], as well as development of pulmonary edema following intratracheal injection of bile acid solutions [99]. Furthermore, Zhou et al. demonstrated increased susceptibility to RDS by impaired uteroplacental blood flow using a rat model with experimentally induced cholestasis [100].

To stratify neonatal risk prospectively, Zecca and De Luca proposed, in 2008, a predictive score for neonatal RDS, integrating maternal serum bile acid concentration, duration of exposure and gestational age at delivery, calculated as follows:

RDS risk score = [maternal bile acids within 24 h of delivery (μmol/L) × exposure time (number of days between ICP diagnosis and delivery)]/gestational age at delivery (days)

A score equal to or greater than 9 identified infants at high risk of RDS, with sensitivity of 85% and specificity of 87%, supporting its clinical utility in decision-making regarding antenatal corticosteroids and mode/timing of delivery [101,102].

### 5.6. NICU Admission and Other Complications

Finally, infants born to mothers with intrahepatic cholestasis of pregnancy exhibit a consistently increased risk of neonatal intensive care unit (NICU) admission, particularly in the setting of moderate-to-severe disease. Recent meta-analyses demonstrate that neonates of ICP pregnancies have more than double the odds of requiring NICU admission compared to controls, with the most common indications for admission including respiratory distress, prematurity-related complications and the need for close monitoring in cases of meconium-stained amniotic fluid [62,103]. In contrast, data regarding low Apgar scores and prolonged neonatal hospitalization remain inconclusive. While some observational studies suggest a trend toward lower Apgar scores in ICP-exposed neonates, these differences are not consistently significant after adjusting for gestational age and mode of delivery [104]. Therefore, NICU admission stands as the most robust and reproducible marker of neonatal morbidity in the context of ICP and should be anticipated/managed accordingly in high-risk pregnancies.

Emerging evidence further supports the long-term implications of these early morbidities. A recent large-scale Swedish population-based cohort found that offspring exposed to ICP had higher odds of neurodevelopmental diagnoses, including attention deficit/hyperactivity disorder, autism spectrum disorder and intellectual disability, with the risk being particularly elevated when ICP was diagnosed early in pregnancy (before 28 weeks’ gestation). Importantly, these associations persisted in comparisons with siblings and maternal cousins, suggesting that the findings are unlikely to be explained by shared familial or genetic factors. While the observational nature of the study precludes causal inference, the results highlight a potential biological link between maternal cholestasis and later neurodevelopmental vulnerability. Notably, the analysis did not include maternal bile acid concentrations and was conducted before ursodeoxycholic acid was widely introduced in Sweden, raising the possibility that therapeutic modulation of bile acid levels may influence these outcomes [6].

For better clarity, Table 1 summarizes the neonatal outcomes associated with intrahepatic cholestasis of pregnancy, outlining both the current evidence and the existing limitations.

## 6. Preventive Strategies

Despite decades of research, no pharmacologic or lifestyle intervention has been shown to prevent the onset of intrahepatic cholestasis of pregnancy [4]. The condition arises in genetically susceptible women when the physiological hormonal and metabolic changes of late gestation surpass the adaptive capacity of bile acid transport and clearance, making universal prevention unlikely. Large clinical trials and systematic reviews consistently demonstrate that agents used after diagnosis, such as ursodeoxycholic acid, improve symptoms and biochemical indices but do not alter the risk of disease development [105]. Nevertheless, certain measures may help reduce risk or facilitate early intervention. Women with a history of ICP, a positive family history or known variants in transport genes such as *ABCB4* or *ABCB11* can benefit from closer surveillance in subsequent pregnancies, allowing for earlier diagnosis and timely therapy [106]. While adequate maternal nutrition remains essential for overall pregnancy health, studies of vitamin K, selenium and other micronutrients indicate associations with ICP, but have not provided evidence that supplementation prevents its occurrence [107]. Supportive strategies such as skin hydration, cool baths and avoidance of heat may help alleviate pruritus once the disease is established, yet they cannot avert its onset [56].

Taken together, the current best practice emphasizes vigilance rather than prevention: identifying women at risk, maintaining awareness of early symptoms such as pruritus and ensuring timely investigation. Until novel mechanistic insights translate into targeted interventions, prevention of ICP remains an unmet clinical need [32].

## 7. Therapeutic Approaches

While intrahepatic cholestasis of pregnancy is primarily managed to alleviate maternal symptoms and prevent fetal complications, therapeutic strategies have increasingly been evaluated for their potential to influence neonatal outcomes. Ursodeoxycholic acid (UDCA) remains the most widely prescribed pharmacologic agent due to its established efficacy in reducing maternal pruritus and improving biochemical markers of cholestasis. Its primary mechanism of action is to reduce the proportion of hydrophobic, hepatotoxic bile acids in the maternal circulation (e.g., chenodeoxycholic acid, taurocholic acid) by replacing them with their more hydrophilic forms, thereby lowering overall bile acid toxicity. Additionally, UDCA promotes bile acid efflux into the canalicular space by enhancing the activity of the bile salt export pump and restores bile acid FXR-regulation to its physiological state [108,109]. More importantly, ursodeoxycholic acid appears to relieve pruritus by reducing autotaxin activity and lowering the levels of circulating lysophosphatidic acid, two pruritogenic molecules implicated as mediators of the cholestatic itch [110]. Recent studies have also demonstrated that UDCA significantly reduces both serum and urinary concentrations of sulfated progesterone metabolites, an effect that closely correlates with the improvement in pruritus in ICP [23].

However, despite widespread use, high-quality evidence demonstrating a direct benefit on perinatal outcomes remains inconclusive. A recent double-blind, randomized, placebo-controlled trial found no statistically significant reduction in perinatal death, preterm birth or NICU admission with UDCA compared to placebo, findings which were reinforced by the largest individual participant data meta-analysis to date, encompassing over 6000 pregnancies [11,111]. Nonetheless, ursodeoxycholic acid continues to be recommended as first-line therapy, with a starting dose of 10–15 mg/kg/day divided in 2 to 3 doses, particularly in women with moderate-to-severe disease, due to its favorable safety profile and maternal symptom control [8].

In refractory or severe cases, particularly where serum bile acid concentrations exceed 100 μmol/L or maternal symptoms persist despite UDCA administration, adjunctive therapies have been investigated. Rifampin, a pregnane X receptor agonist, may enhance bile acid clearance by upregulating hepatic enzymes and transporters involved in bile acid metabolism. This broad activation of hepatic detoxification pathways is thought to accelerate the elimination of various pruritogenic substances, including bile acids and lysophosphatidic acid, thereby contributing to symptomatic relief [112,113]. Although limited to case series and small observational cohorts, rifampin co-administration has demonstrated additional reductions in bile acids when added to UDCA, with some evidence of improved maternal well-being. However, concerns regarding hepatotoxicity and insufficient neonatal data currently limit its use to select cases under close monitoring [114,115]. Its role in ICP management is being further evaluated in the ongoing TURRIFIC trial, a multicenter randomized controlled study comparing rifampin and UDCA in women with severe cholestasis, with results expected in 2026 [116]. Other experimental agents, including S-adenosylmethionine (SAMe), cholestyramine, guar gum and active charcoal, have been studied with inconsistent results and are not routinely recommended due to limited efficacy and potential side effects, such as fat-soluble vitamin depletion [74,105].

Beyond pharmacologic therapy, timing of delivery remains the most critical intervention influencing neonatal prognosis. There is robust epidemiological and mechanistic evidence linking bile acid levels ≥ 100 μmol/L to an increased risk of late intrauterine fetal demise. As such, most clinical guidelines recommend planned delivery between 36 and 37 weeks in pregnancies complicated by severe ICP or earlier if additional risk factors such as fetal distress, twin gestation or preeclampsia are present. For women with mild disease (bile acids < 40 μmol/L), expectant management until 38 weeks may be appropriate, provided close biochemical and fetal surveillance is maintained. This risk-stratified approach enables clinicians to mitigate the risk of stillbirth while minimizing the iatrogenic consequences of late-preterm or early-term birth [4,77,117].

## 8. Future Directions

While pharmacologic agents and delivery planning form the current cornerstone of ICP management, there is growing interest in advancing fetal monitoring strategies and exploring novel therapeutic targets. Several studies have identified fetal cardiac involvement in ICP, including altered heart rate variability, supraventricular arrhythmias and elevated fetal NT-proBNP levels, especially in the setting of severe maternal hypercholanemia. Echocardiographic assessments, including the modified myocardial performance index (mMPI), have been proposed as potential tools to detect subclinical myocardial dysfunction. However, these modalities require validation in larger cohorts and are not yet part of standard prenatal surveillance protocols [78,81,82].

From a therapeutic development standpoint, obeticholic acid and norucholic acid (norUDCA) are being investigated for their ability to restore bile acid homeostasis and reduce systemic toxicity. Preclinical studies suggest enhanced hepatobiliary transport and anti-inflammatory activity, but human data in pregnancy are lacking and potential teratogenicity remains a concern [118,119]. Additionally, machine learning models incorporating maternal bile acid trajectories, gestational age and dynamic clinical parameters are under development to personalize delivery decisions, but these algorithms are in early-stage validation [120].

Ultimately, the optimal management of ICP will likely involve a multimodal approach combining biochemical control, individualized fetal surveillance and delivery timing tailored to maternal and fetal risk profiles. Further randomized controlled trials are urgently needed to evaluate emerging therapies, define precise bile acid thresholds for intervention, and establish validated tools for fetal monitoring that move beyond gestational-age-based strategies.

## 9. Conclusions

Intrahepatic cholestasis of pregnancy is a well-recognized gestational liver disorder with significant implications for neonatal health. Elevated maternal bile acid levels are consistently associated with a spectrum of neonatal complications, including preterm birth, meconium-stained amniotic fluid, neonatal respiratory distress syndrome and, in severe cases, intrauterine fetal demise, with the greatest risks observed at serum concentrations ≥ 100 μmol/L. Current management strategies are centered on maternal symptom relief and fetal protection through biochemical monitoring, pharmacologic therapy and timely delivery. Ursodeoxycholic acid remains the most widely used agent, with proven benefits in maternal pruritus and liver biochemistry, although definitive neonatal benefit has not yet been established.

Despite advances in understanding the pathophysiology and clinical course of ICP, several areas remain inadequately addressed. Standardized fetal surveillance strategies are lacking, while promising markers such as fetal myocardial performance index and NT-proBNP require validation. Adjunctive therapies such as rifampin may offer promise in refractory cases, though safety data in neonates remain limited. Furthermore, the long-term outcomes of neonates exposed to elevated bile acid levels in utero are not well defined and should be a focus of prospective follow-up studies.

Improving neonatal outcomes in ICP will require individualized management that integrates maternal disease severity, dynamic biochemical trends and fetal status. Ongoing research should prioritize high-quality randomized trials, validation of predictive models, and mechanistic studies to better characterize fetal vulnerability and therapeutic response. Until then, clinical care should emphasize risk stratification, evidence-based timing of delivery, and multidisciplinary coordination to optimize both maternal and neonatal outcomes.

## Figures and Tables

**Figure 1 biomedicines-13-02066-f001:**
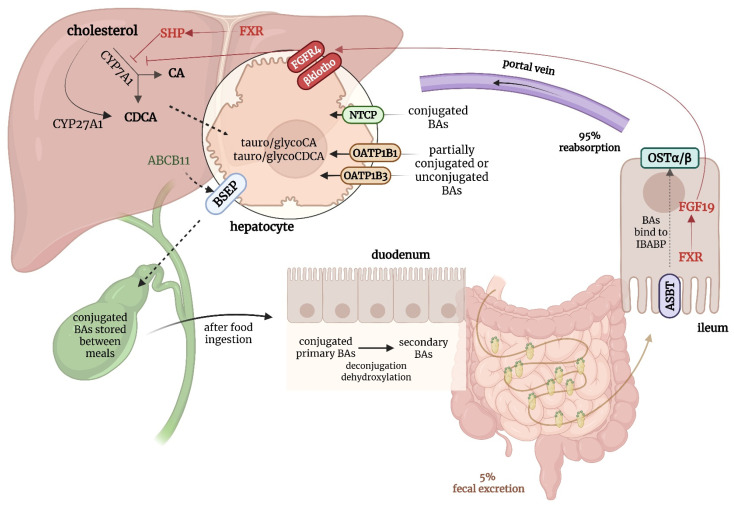
Enterohepatic circulation of bile acids and its regulation. Figure created using BioRender.

**Figure 2 biomedicines-13-02066-f002:**
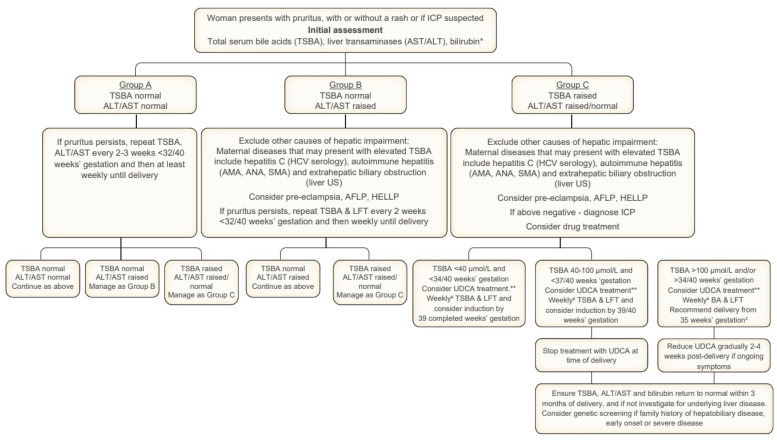
Intrahepatic cholestasis of pregnancy management algorithm. This flowchart summarizes the evaluation of pregnant women with suspected ICP, beginning with pruritus and initial laboratory testing. Patients are stratified into three groups: Group A with normal bile acids and liver tests (monitor and repeat testing); Group B with normal bile acids but raised liver tests (exclude other causes and monitor); Group C with raised bile acids (diagnose ICP and guide management by bile acid thresholds). The thresholds of <40, 40–100 and >100 μmol/L direct pharmacological therapy and decisions regarding timing of delivery. * Bilirubin concentrations are rarely raised in ICP and if marked or persistent, investigations should be performed to identify the cause. ^¥^ UTSBA should be checked at least weekly as they may continue to rise with advancing gestation. ** UDCA protects against spontaneous preterm birth in singleton pregnancy and may also protect against stillbirth. ‡ The risk of stillbirth rises markedly from 35 weeks’ gestation in women with TSBA > 100 μmol/L. AFLP, acute fatty liver of pregnancy; LFT, liver function tests; ALT, alanine aminotransferase; AMA, anti-mitochondrial antibody; ANA, antinuclear antibody; AST, aspartate aminotransferase; HCV, hepatitis C virus; HELLP, hemolysis elevated liver enzymes and low platelets; ICP; intrahepatic cholestasis of pregnancy; SMA, smooth muscle antibody; TSBA, total serum bile acids; US, ultrasound; UDCA, ursodeoxycholic acid. Reproduced with permission from the work of Williamson et al. [56].

**Table 1 biomedicines-13-02066-t001:** Summary of evidence on neonatal outcomes associated with ICP.

Neonatal Outcome	Evidence	Association with Bile Acids	Limitations/Controversies
Preterm birth	Increased risk (spontaneous and iatrogenic); dose–response relationship	Strong correlation, especially ≥40 μmol/L	Confounding by delivery policies
Meconium-stained amniotic fluid	15–48% of ICP pregnancies	Strongest with bile acids ≥ 100 μmol/L	Mechanism mostly inferred from animal studies
Stillbirth	Markedly elevated when ≥100 μmol/L (10-fold increase)	Clear association	Unpredictable, sudden onset; limited preventive markers
Low birth weight/ Small for gestational age	Conflicting evidence (↑ risk in Asian cohorts, not consistent in Western populations)	Possibly indirect via prematurity	Heterogeneity; guidelines do not support routine growth monitoring
Large for gestational age	Some reports in Scandinavian cohorts	Mechanism unclear (metabolic?)	Contradictory to SGA data
Neonatal RDS	17–29% prevalence even after correction for gestational age	Bile acid toxicity, surfactant dysfunction	Limited prospective studies
NICU admission	>2-fold increased risk, mostly due to prematurity and RDS	Consistently reported	Robust finding, but not specific to ICP
Apgar score	Some studies suggest lower scores at 1 and 5 min	Not consistently linked to bile acid levels	Inconsistent; differences often disappear after adjustment
Length of hospital stay	Some reports of prolonged hospitalization in ICP-exposed neonates	Not directly associated	Data limited; often confounded by prematurity and NICU admission
Long-term outcomes	Higher risk of ADHD, autism, intellectual disability (Swedish data)	Not directly linked to bile acids in current studies	Observational only; no causal inference

Legend: ICP = intrahepatic cholestasis of pregnancy; SGA = small for gestational age; RDS = respiratory distress syndrome; NICU = neonatal intensive care unit; ADHD = attention deficit/hyperactivity disorder.

## Data Availability

Not applicable.

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
