# Peer review of "Intrahepatic Cholestasis of Pregnancy: Neonatal Impact Through the Lens of Current Evidence"

_biomedicines, 2025, doi:10.3390/biomedicines13092066_

Round 1
Reviewer 1 Report
Comments and Suggestions for Authors
Dear authors;
Thank you for your valuable manuscript, and I really enjoyed reading it. Below I provide my commentaries:
Abstract:
- Please capitalize the first letter of each keyword.
Introduction:
- Line 40, “about” seventy years ago. There are two different years.
- Please add a space before the references.
- Line 56, it would be better to use more recent references.
- Line 59, reference number 7 is not related to your sentence. Please review it.
Clinical Presentation and Diagnosis:
- Line 259, as indicated in the reference, it should be noted that physiological changes during pregnancy may lead to elevated levels. Additionally, the test is performed in a non-fasting state due to pregnancy conditions.
Low birth weight and small for gestational age:
- Please add reference for line 424-430.
- Line 476, these two articles were published in completely different years and with groups of authors. Please revise this section accordingly.
Reviewer 2 Report
Comments and Suggestions for Authors
This review, “Intrahepatic Cholestasis of Pregnancy: Neonatal Impact Through the Lens of Current Evidence” by Lucia et al., is a well-structured and informative narrative review. The figures are clear and relevant, and the manuscript follows a logical flow. The language is smooth and accessible, making it suitable for both clinical and academic audiences.
That said, I have a few minor suggestions that may help further improve clarity and balance:
1-Since this is a narrative review, a brief explanation of how the literature was selected (e.g., prioritizing recent guidelines, large-scale studies, or mechanistic work) would help clarify the scope.
2-While the sections on fetal demise and respiratory distress are well developed, others—such as NICU admissions and long-term neonatal outcomes—are comparatively brief. Slightly expanding these areas may improve overall balance.
3-The conclusion is somewhat repetitive. Streamlining it could help emphasize the key messages more clearly.
4-Figure 2 is informative but visually dense. Consider simplifying the layout or splitting it into two separate visuals for clarity.
Reviewer 3 Report
Comments and Suggestions for Authors
Dear Authors,
I reviewed your review manuscript, entitled 'Intrahepatic Cholestasis of Pregnancy: Neonatal Impacts through the Lens of Current Evidence,' with great interest. The narrative review provides information regarding pathophysiological mechanisms, clinical impacts, and management strategies related to neonatal outcomes in ICP. The manuscript contains figures with proper design and gives constructive information to the readers.
The manuscript has some concerns that need to be considered by the author to enhance its quality.
- It is suggested to discuss the role of hormones, particularly estrogen and progesterone, as a risk factor for ICP and their mechanism of action.
- It is also suggested to add information regarding the role of the placenta and its relation to ICP, the involved mechanism of action.
- It is also suggested to add some information regarding the microbiome and its association with ICP.
- Adding some information or a section on the preventive strategies will enhance the quality of the review manuscript.
- Some grammar errors need to be revised.
Round 2
Reviewer 3 Report
Comments and Suggestions for Authors
Dear authors,
I received the revised version of the manuscript. The authors have considered the suggestions and have made some changes to the manuscript accordingly. The manuscript in its current version is acceptable for publication in the journal.
Regards,